# Processing and Properties of ZrB_2_-Copper Matrix Composites Produced by Ball Milling and Spark Plasma Sintering

**DOI:** 10.3390/ma16237455

**Published:** 2023-11-30

**Authors:** Iwona Sulima, Grzegorz Boczkal

**Affiliations:** 1Institute of Technology, University of the National Education Commission, Krakow Podchorazych 2 St., 30-084 Krakow, Poland; 2Faculty of Non-Ferrous Metals, AGH University of Krakow, Mickiewicza 30 Av., 30-059 Krakow, Poland; gboczkal@agh.edu.pl

**Keywords:** copper matrix composites, diboride zirconium (Orb), powders, mechanical milling, Spark Plasma Sintering (SPS)

## Abstract

Copper matrix composites with zirconium diboride (ZrB_2_) were synthesised by ball milling and consolidated by Spark Plasma Sintering (SPS). Characterisations of the ball-milled composite powders were performed by scanning electron microscopy (SEM), X-ray diffraction, and measurement of the particle size distribution. The effect of the sintering temperature (1123 K, 1173 K, and 1223 K) and pressure (20 MPa and 35 MPa) on the density, porosity, and Young’s modulus was investigated. The relationship between the change of Orb content and physical, mechanical, and electrical properties was studied. Experimental data showed that the properties of Cu–Orb composites depended significantly on the SPS sintering conditions. The optimal sintering temperature was 1223 K with a pressure of 35 MPa. Composites exhibited a high degree of consolidation. For these materials, the apparent density was in the range of 93–97%. The results showed that the higher content of Orb in the copper matrix was responsible for the improvement in Young’s modulus and hardness with the reduction of the conductivity of sintered composites. The results showed that Young’s modulus and the hardness of the Cu 20% Orb composites were the highest, and were 165 GPa and 174 HV0.3, respectively. These composites had the lowest relative electrical conductivity of 17%.

## 1. Introduction

Copper and its alloys are widely used because of their excellent electrical, thermal, and mechanical properties, among others, in the aviation, railway, military, and electronics industries [1]. At the same time, the intensive development of modern technologies places increasing demands on engineering materials. This requires the use of various methods to improve the properties of copper alloys. One of them is the reinforcement of the copper matrix by introducing hard ceramic phases [2,3]. The result is an improvement in mechanical and physical properties such as strength, hardness, creep resistance, and thermal conductivity. The improvement in the mechanical properties of Cu composites depends on the properties of the reinforcement phase and the interface between the reinforcement and copper [4]. Ceramics such as oxides [5,6], carbides [7,8] and borides [9,10] are good candidates for the reinforcement of copper-based composites. The addition of hard ceramic particles to the soft copper matrix significantly improves its strength and resistance to high temperatures, as well as its resistance to wear, without significant deterioration in thermal and electrical conductivity [11,12]. Of particular interest are studies on improving electrical [13,14], thermal [15,16], and tribological [17] properties, which aim at obtaining Cu composites with wide practical applications. Cu-based composites are used in the automotive, military, electrical, and electronic industries, and in heat exchangers in power plants [7,10,15,18]. According to the literature, transition-metal borides have significantly higher electrical conductivity, lower thermal expansion, and better wettability of molten copper compared to oxides and carbides. Zirconium diboride is characterised by a high melting point (>3273 K), high hardness (22–25 GPa), elastic modulus (440-460 GPa), good wear and oxidation resistance, and excellent thermal and electrical properties [19,20,21]. Therefore, zirconium diboride is increasingly used as a reinforcement phase in composites with matrix, including copper [22,23], aluminium [24,25], nickel [26], titanium [27], and their alloys.

Pure copper (Cu) is characterised by high electrical and thermal conductivity and good corrosion resistance, but unfortunately low strength and wear resistance [28]. Orb is a potential candidate for reinforcing copper, improving its strength and wear resistance [22,23,29,30]. Table 1 presents selected literature data [23,31,32,33,34] on the influence of the ceramic reinforcing phase on the properties of Cu-based composites. Fan et al. [22] prepared by the melting–casting method of Cu-ZrB_2_ composites. They showed that as the Orb content increased, the hardness and abrasion resistance of the composites increased. At the same time, the electrical conductivity of the composites decreased. In turn, Wang et al. [23] produced Cu–Orb composites by hot sintering. They showed that the relative density and electrical conductivity of the composites decreased with increasing ZrB_2_ content (1–9 wt%).

However, the microhardness reached a maximum value of 100.8 HV0.2 when the ZrB_2_ content increased to 7 wt% and then decreased. In another work [30], Shaik and Golla investigated the effect of ZrB_2_ (1, 3, 5, 10 wt%) on the mechanical properties and abrasive wear of copper. An improvement in wear resistance, hardness, and compressive strength was observed with an increase in the amount of reinforcement phase in the copper matrix.

Spark plasma sintering (SPS) is widely used to consolidate various materials such as metals [35,36], cermets [37,38], ceramics [39,40], and composites [41,42,43,44]. In the SPS process, the electric pulse current flows directly through sintered powder materials, which can generate very high heating and cooling rates. This promotes evaporation, cleaning, and activation of the surface of powder particles and improves the diffusion mechanism. In addition, it allows you to control the grain growth process. SPS technology allows for a reduction of the sintering temperature and a shortening of the sintering time. The advantage of this technique compared to conventional sintering is the consolidation of materials without the need for preliminary compaction [45,46,47,48].

In the present study, SPS technology was used for the consolidation of the Cu composites reinforced with ZrB_2_. The first stage of the research focused on the characteristics of composite powder mixtures with various ZrB_2_ contents prepared by milling in a high-energy ball mill. Then the influence of the sintering conditions (temperature and pressure) and the content of the ZrB_2_ reinforcing phase (5 wt%, 10 wt%, 15 wt% and 20 wt%) on the physical, mechanical, and electrical properties of the composites were examined.

## 2. Experimental Procedures

### 2.1. Raw Materials

The composite materials were made of copper powders (10 μm, 99.9 wt% purity, Kamb Import Export, Warsaw, Poland) which comprised the matrix material and zirconium diboride powder (2.5–5.5 μm, 99.9 wt% purity, H.C. Starck Tungsten GmbH, Goslar, Germany) constituting the reinforcement of the composite. Figure 1 and Figure 2 present the morphology of the starting powders and the results of the particle size tests. Measurements of the particle size distribution of the starting powders and composite mixtures were performed in polypropylene alcohol using the SALD-7500nano analyser (Shimadzu Corporation, Kyoto, Japan) with WingSALD II software (Version 3.4.9), which allows the automatic calculation of the refractive index and tracking changes in the particle size distribution in real time. Measurements were carried out with a measurement step of 1 s. Before each test, the powders immersed in alcohol were broken up in an ultrasonic bath for 5 min.

### 2.2. Fabrication of the Cu–ZrB_2_ Composites

The mechanical milling of the composite mixtures was performed in high-energy planetary ball mill Pulverisette 4 (Fritsch GmbH, Idar-Oberstein, Germany). Four composite mixtures were prepared with the following composition: Cu + 5 wt% ZrB_2_, Cu + 10 wt% ZrB_2_, Cu + 15 wt% ZrB_2_, and Cu 20 wt% ZrB_2_. The powders were milled using the following cycle: 20 min of milling—10 min break. The balls/powder weight ratio was 1:5 (50 g: 250 g). The container and balls used were made of tungsten carbide. Table 2 shows the parameters of the milling process of composite powders in a high-energy mill. Stearic acid was used as the process control agent. The milling operation was interrupted periodically after milling for 5, 10, 15, and 20 h to determine the change in the morphology and phase composition of the powders during milling. Furthermore, after the milling process, the morphology of the powders was characterised using a scanning electron microscope (SEM JSM 6610LV, Tokyo, Japan); the particle size distribution of the powders (SALD-7500 nano analyser, Shimadzu Corporation, Kyoto, Japan), and the phase composition were characterised using an X-ray diffractometer (XRD, Malvern Panalytical, Almelo, Netherlands).

The powders were sintered by SPS (LSP-100, Laboratory Sintering Press Dr Fritsch GmbH, Fellbach, Germany). The sintering process was carried out in an argon atmosphere under a pressure of 20 MPa and 35 Mpa for 5 min. To explore the effect of sintering temperature on the properties of the composites, the sintering temperature was set at 1123 K, 1173 K, and 1223 K. The furnace heating rate was kept at 473 K/min. The samples were cooled at a rate of 373 K/min. The powder mixtures were put into a cylindrical graphite die with an inner diameter of 20 mm. After the sintering process, the samples had a height of 7–8 mm. During the SPS process, the temperature was monitored with a pyrometer. Figure 3 shows the selected sintering parameters that were registered during the SPS process, as a function of the process time: temperature, force, and punch displacement.

### 2.3. Material Characterisation

The density of the sintered composites was measured by Archimedes’ principle. Relative density is the ratio of measured density to theoretical density. An analytical balance RADWAG AS 220/C/2 (Radwag, Radom, Poland) was used to measure the weight of the samples. Young’s modulus was determined using the Panametrics Epoch III flaw detector (Panametrics, Billerica, MA, USA). Young’s modulus was measured based on the velocity of ultrasonic waves transition through the sinter. The velocities of the transverse and longitudinal waves were calculated as the ratio of sample thickness to relevant transition time. For each sample, five measurements of Young’s modulus were performed. The measurement error was 2%.

The NEXUS 4000 microhardness tester (Innovatest EuropeBV, Maastricht, The Netherlands) was used to measure the average hardness of the composites at the load of 2.942 N and a hold time of 10 s. Each sample was tested with 10 points.

X-ray diffraction analysis (XRD) was carried out using a diffractometer equipped with a Cu/Kα radiation. The microstructures of the sintered composites were evaluated using scanning electron microscopy (JEOL JSM 6610LV, Tokyo, Japan) equipped with energy-dispersive spectroscopy (EDS, Aztec, Oxford Instruments, High Wycombe, UK).

The electrical conductivity of the samples was determined according to the diagram in Figure 4 using the four-point method [49,50]. The measurement system used a 2182A 2-channel nanovoltmeter (Tektronix UK Ltd., Berkshire, UK) and a precision Array Electronic 3644A programmable power supply 18 V/5 A (Array Electronic Co. Ltd., Nanjing, China). The current flowing in the measurement circuit did not exceed 0.5 A. This allowed us to avoid heating of the sample during measurement. Each single measurement was performed as the average of two results obtained with different polarisations. Five measurements were made in each sample variant and the results were averaged.

## 3. Research Results

### 3.1. Characterisation of Composite Powder Mixtures

The changes in the shape and size of the powder particles that occurred during milling were analysed using a scanning electron microscope and a nano analyser. The microstructures and particle size distributions of composite powders containing 5 wt% ZrB_2_ milled for 5, 10, 15, and 20 h, respectively, are shown in Figure 5, Figure 6, Figure 7 and Figure 8. The starting copper powder was characterised by a spherical shape and an average particle size of 10 μm (Figure 1). The ZrB_2_ powder particles were shaped like polyhedra with sharp edges (Figure 2). The use of the milling process resulted in a change in the morphology and size of the composite powders. After 5 h of milling (Figure 5), the composite powder particles flattened. There was an increase in particle size (Figure 5b, Table 3) to approximately 17 µm. In the initial stage of the milling process, powder particles were subjected to high-energy collisions, which caused plastic deformation. This led to a strengthening of the particles and subsequent cracking. As a result of cracking, new surfaces were created, which allowed the powder particles to bond with each other [51,52]. The consequence of the bonding process was the growth of the composite powder particles in relation to the particle size of the initial copper powder (Figure 5). In the subsequent stages of milling (after 10, 15, and 20 h), the matrix powder particles were further deformed (Figure 6, Figure 7 and Figure 8), and hard reinforcement particles became fragmented due to severe plastic deformation. As a result, the particle size of the composite powder gradually decreased as a function of the milling time (Table 3). After 20 h of milling, the average particle size of the composite powder with 5 wt% ZrB_2_ was approximately 13 μm (Figure 8). The powder particles had a flake shape with a developed and wrinkled surface. Analogous changes in the shape of the powders were observed as a function of the milling time (5–20 h) for other composite mixtures containing 10, 15, and 20 wt% ZrB_2_.

Furthermore, the results of testing the composite mixtures (Figure 9) showed that changing the amount of the reinforcement phase had a beneficial effect on the particle size distribution. After 20 h of milling, a decrease in the average particle size of the powders was observed with an increase in the amount of ZrB_2_. For comparison, the average particle size of the composite powder was approximately 13 μm, 11.5 μm, 7 μm, and 5 μm for contents of 5, 10, 15, and 20 wt% ZrB_2_, respectively (Table 4). During the milling process, the Cu powder tended to cold-weld together, while the brittle and hard ZrB_2_ particles would fragment. In this case, zirconium diboride can act as a milling aid, especially when its amount in the Cu matrix increases [53]. Figure 10 shows a comparison of the results of X-ray analyses of composite powders containing 5% ZrB_2_ after selected milling times. The diffraction peaks of copper and ZrB_2_ are clearly visible in the X-ray recording of the composite powders after each milling stage (5–10–15–20 h). They also revealed the presence of small amounts of CuZr and Zr phases in the powder mixtures. Similar test results were obtained for all composite powders. No changes in the intensity of the peaks originating from Cu, ZrB_2_, Zr, and CuZr were observed in the diffractograms as the milling time increased.

### 3.2. Characterisation of Sintered Cu–ZrB_2_ Composites

The parameters of the SPS process play a significant role in shaping the properties of the sintered materials. Therefore, in the first stage of the research, optimisation of the sintering conditions was carried out. Sintered Cu–ZrB_2_ composites were evaluated in terms of the degree of compaction (density and porosity) to indicate optimal sintering conditions. An additional parameter taken into account in the optimisation of the sintering conditions was the value of Young’s modulus determined using the ultrasonic method. The results of the tests on the influence of the sintering pressure and temperature on the properties analysed are presented in Table 5 and Table 6. Composites sintered at temperatures of 1123 K, 1173 K, and 1223 K at a pressure of 20 MPa were characterised by low relative density (84–89%), high porosity (11.34–6.95%) and low Young’s modulus values (88–106 GPa). The high porosity of the sintered material indicates the presence of defects and inhomogeneities in the material, which may affect the attenuation of ultrasonic waves during measurements and obtain low values of Young’s modulus [54,55]. Increasing the sintering pressure to 35 MPa resulted in a high degree of densification. The relative density of the sintered composites was in the range of 95–97% (Table 5), and the open porosity decreased to 1.12–2.54%. Similarly, higher values of Young’s modulus were obtained. The test results showed (Table 6) that very similar values of density, open porosity, and Young’s modulus were determined for copper without a reinforcing phase, which was sintered at temperatures of 1123–1223 K. These results recommend performing the SPS process for pure copper at a lower temperature of 1123 K. However, in the case of Cu–ZrB_2_ composites, the temperatures of 1123 K and 1173 K are insufficient to obtain a high degree of densification. The negative effect of these sintering conditions was particularly clearly observed for Table 6 (the influence of the sintering pressure on the properties of the materials formed by SPS).

Meanwhile, increasing the temperature of the SPS process to 1223 K improved the density of the composites and porosity, and increased Young’s modulus. For this sintering temperature, the highest values of relative density above 90%, low porosity, and a higher Young’s modulus in the range (92–96%) were achieved. Increasing the sintering temperature is beneficial because a higher temperature accelerates atomic diffusion and increases the migration rate of grain boundaries, which in turn promotes a reduction of pore size and an improvement in density [56,57]. This is consistent with the results of other works [33,58,59]. Wang et al. [33] investigated the effect of the SPS process temperature at a pressure of 40 MPa on the properties of Cu–TiC composites. They showed that increasing the temperature of the SPS process improved the relative density of sintered materials. Soloviova et al. [58] demonstrated similar correlations for Cu–(LaB_6_-TiB_2_) composites also prepared using the SPS method. 

Analysis of the results indicated a characteristic dependence of the properties of sintered composites as a function of the change in the amount of the ZrB_2_-reinforcing phase (Figure 11 and Figure 12, Table 6). The apparent density of the composites decreased from 8.52 g/cm^3^ to 7.57 g/cm^3^ with an increase in the content of ZrB_2_ (5–20%). The reduction in the density of the composites is mainly due to the lower density of ZrB_2_ ceramics (6.10 g/cm^3^) compared to copper (8.96 g/cm^3^) [28,60,61]. Wang et al. [23] observed a similar trend in their research. They produced Cu–ZrB_2_ composites using a hot-pressing sintering process at a temperature of 840 °C and a pressure of 25 MPa. A decrease in density was found (from 96.1 to 91.3%) when the ZrB_2_ content was changed (1–9 wt%). The test results showed that the Young’s modulus increased with the increase in the ZrB_2_ content in the copper matrix (Figure 12). The values of Young’s modulus were 105 GPa, 123 GPa, 139 GPa, 154 GPa, and 165 GPa (Table 6) for sintered copper and composites with 5 wt%, 10 wt%, 15 wt%, and 20 wt% ZrB_2_, respectively. The introduction of 20 wt% ZrB_2_ into the copper matrix resulted in an increase in the value of Young’s modulus by approximately 60% compared to sintered copper without a reinforcing phase. Hardness also showed a positive correlation with the change in ZrB_2_ content (Figure 13). Furthermore, only the addition of 5 wt% ZrB_2_ increased the hardness twice compared to pure copper. Increasing the ZrB_2_ content had a beneficial effect on improving hardness. Composites containing 20 wt% ZrB_2_ were characterised by 3x higher hardness (174 HV0.3) compared to sintered copper (61 HV0.3). This is the result of the presence of the ZrB_2_ reinforcement phase in the copper matrix, which has a very high hardness (>2200 HV [58]). The improvement of microhardness can be explained by the uniform distribution of ZrB_2_ particles within the Cu matrix. Wang et.al. [23] reported that the microhardness of the pure copper (57.5 HV0.2) increased to 100.8 HV0.2 with increasing ZrB_2_ content up to 7 wt%.

Microstructural studies with chemical composition (EDS) and phase composition (X-ray diffraction) analyses were carried out for all sintered Cu–ZrB_2_ composites. For comparison, Figure 14 and Figure 15 show exemplary microstructures of Cu–10 wt% ZrB_2_ and Cu–20 wt% ZrB_2_ composites. A uniform distribution of the reinforcing phase was observed in the copper matrix. Only for composites containing 20 wt% ZrB_2_, the local formation of ZrB_2_ agglomerates was demonstrated (Figure 15). Phase analyses (Figure 16) confirmed the presence of only the ZrB_2_ and copper phase for all sintered composites.

Comparisons of relative electrical conductivity for sintered materials are presented in Figure 17. The results obtained were related to the conductivity of pure electrolytic copper ([62]), whose value was taken as 100%. The addition of ZrB_2_ caused a strong decrease in the electrical conductivity of the composites. The effect was most visible with 10% of the content of the reinforcement phase and more. This was caused by a sharp reduction in the share of pure copper in the active cross section of the material. The observed decrease amount and its subsequent stabilisation at a constant level may indicate partial diffusion of the components from zirconium diboride into the copper matrix. According to [63], the solubility of Zr in copper at room temperature is negligible; however, the content of boron dissolved in the structure can be up to 0.06 at% [64]. Such a disturbance at the level of the matrix structure combined with discontinuities resulting from sintering may have a significant impact on the electrical properties and cause the observed effects.

## 4. Discussion

The preparation of composite powders by milling is of key importance for the properties of the material produced in the SPS process. It was observed that during the milling process, the amount of the added reinforcing phase affects the distribution of the particles in the copper matrix. During milling, the higher content of ZrB_2_ in the composite powders improves the uniformity of particle distribution. Moreover, the particle size distribution curve undergoes favourable changes, which tends toward an ideal Gaussian curve without a clear maximum. This effect is especially visible for the content of 5 and 10 wt% ZrB_2_ (Figure 9a,b). Increasing the content of ZrB_2_ in powder mixtures results in the appearance of a significant number of fine particles resulting from the kinetics of the milling process. However, they do not disturb the homogeneous distribution of the reinforcing phase in the matrix.

Furthermore, during the milling process, an initial mechanical synthesis is carried out, resulting in the CuZr phase observed in composite mixtures and a small fraction of pure zirconium particles released from the ZrB_2_ phase. The results of the diffraction analysis (Figure 10) show that the synthesis process of these phases takes place after 5 h of milling. Extending the milling time to 20 h does not affect the formation of subsequent phases, but only affects the number of previously identified ones, as evidenced by the observed changes in the intensity of individual peaks on the graph. The sintered composite material shows a linear dependence of density (Figure 11) and Young’s modulus (Figure 12) on the amount of the reinforcing phase. A similar relationship was observed for microhardness measurements (Figure 13). The linear characteristics of these parameters indicate the homogeneity of the material and allow for good predictions of parameters when using other proportions. The results also show that in the case of the highest ZrB_2_ content (20 wt%), the formation of conglomerates of ZrB_2_ particles can be observed. This is an unfavourable phenomenon and may indicate microstructure homogeneity problems for ZrB_2_ contents of 20 wt% and above. The phase analysis of the sintered composites (Figure 16) showed in all cases the existence of only two phases, i.e., the copper matrix and ZrB_2_. The small amounts of Zr and ZrCu phases observed in powder mixtures after mechanical synthesis disappeared as a result of the diffusion processes accompanying the sintering.

The electrical conductivity of composites containing metal borides is reduced due to changes in the proportions of components with different electrical properties in the cross-section of the sample. In the case of materials with small (short-range) diffusion exchange of components, it should be proportional to the volume fraction of the reinforcing phase. However, for solution hardening materials, where alloy addition occurs evenly throughout the matrix and modifies the parameters of the crystal lattice by introducing more point defects and, consequently, conduction electron scattering centres, even a small amount of the addition may cause a significant decrease in the electrical conductivity of the material.

A similar relationship occurs for thermal conductivity, which has a different nature in terms of the conduction mechanism. However, both mechanisms share common macroscopic features that can be expressed by a general transport equation. They are a generalisation of Ohm’s law for the density of electric current (current flux—*j*) in a conductor:*j* = σE(1)
where: σ—electrical conductivity tensor, E—vector of electric field intensity, and


Fourier’s law describing the amount of thermal energy flow q:q = −κ_x_(dT/dx) (2)
where: κ_x_—thermal conductivity coefficient along the x axis, dT/dx—temperature gradient.

Both E and dT/dx have a gradient character, so in general we can write:*j* = −*β* ∙ *gradA*
(3)
where: *j*—flux density vector of the appropriate amount (size) of internal energy for thermal conduction or charge for electrical conduction, *β*—coefficient of proportionality of thermal or electrical conductivity, *A*—scalar quantity whose gradient causes a given phenomenon (temperature, electric potential) [64].

In the case of the sintered Cu–ZrB_2_ composite, the results indirectly indicate the partial diffusion of boron into the copper matrix, which may explain the observed strong decrease in the electrical conductivity of the samples with the increase in the content of the ZrB_2_ phase. The easy diffusion transfer to the matrix results from the large difference in the atomic radius of copper (140 pm) and boron (85 pm). This means that boron will tend to be located in the interstitial holes of the copper lattice and cause local contraction of the crystal lattice, increasing the scattering potential of conduction electrons. At the same time, as shown by [65], the limit of solubility of boron in copper does not exceed 0.06% at room temperature. A strong initial drop in conductivity and then its stabilisation may indicate that boron has diffused into the matrix and reached limiting solubility.

## 5. Conclusions

A change in SPS temperature from 1123–1223 K led to an increase in the density, Young’s modulus, and hardness. The best physical and mechanical properties were obtained for the composites that were sintered at 1223 K–35 MPa.Furthermore, the influence of the ZrB_2_ content on the microstructure, physical, mechanical, and electrical properties of sintered composites was studied. The results showed that the increase in the ZrB_2_ content in the copper matrix improves Young’s modulus and hardness while reducing the density and electrical conductivity. The addition of the reinforcing phase in an amount greater than 10 wt% negatively affects the properties of the copper matrix composites.The addition of 20 wt% ZrB_2_ causes the beginning of the formation of particle conglomerates and, consequently, disturbs their homogeneous distribution in the copper matrix.A strong decrease in the electrical conductivity of composites containing 5 and 10 wt% ZrB_2_, as well as stabilization of the conductivity in larger amounts, indicates the partial diffusion of boron into the copper matrix.

## Figures and Tables

**Figure 1 materials-16-07455-f001:**
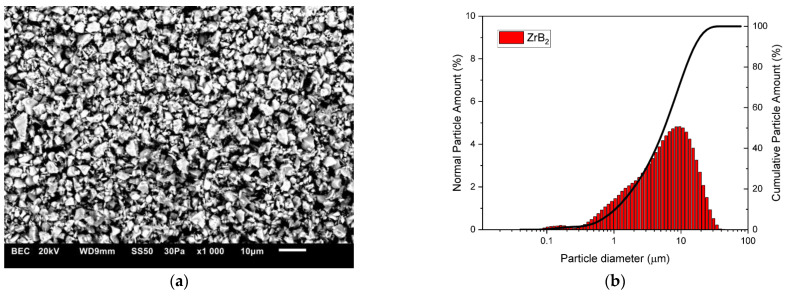
(**a**) Morphology and (**b**) particle size distribution of the ZrB_2_ powder.

**Figure 2 materials-16-07455-f002:**
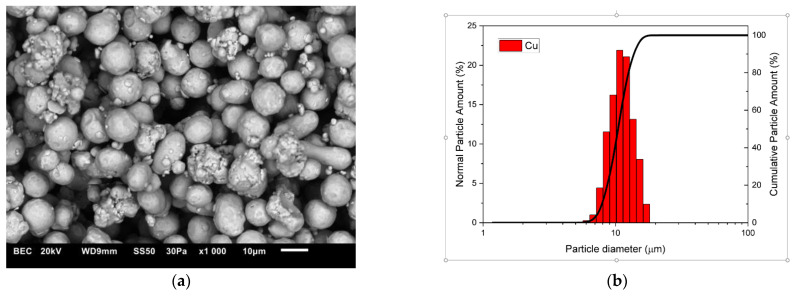
(**a**) Morphology and (**b**) particle size distribution of the Cu powder.

**Figure 3 materials-16-07455-f003:**
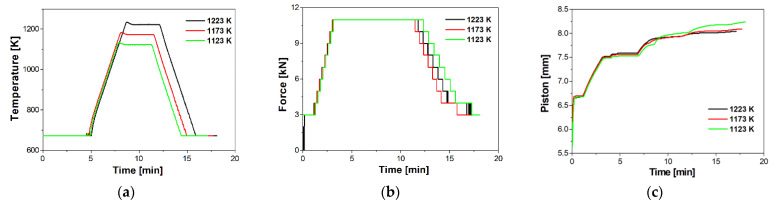
Actual (**a**) temperature-time, (**b**) force-time, and (**c**) punch displacement-time curves registered during SPS.

**Figure 4 materials-16-07455-f004:**
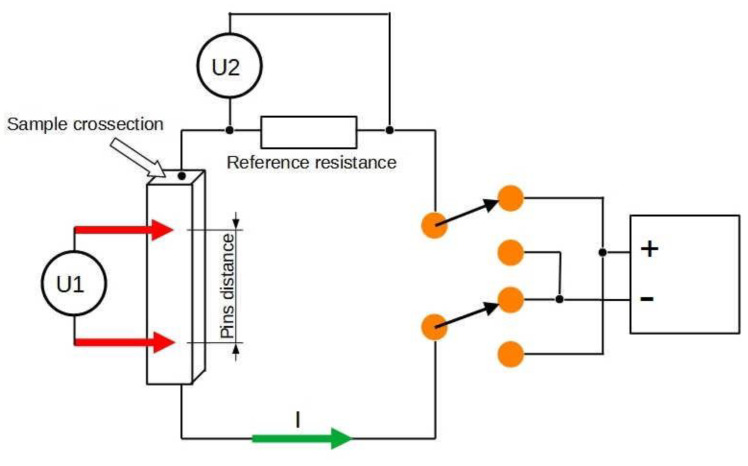
Schematic of the electrical conductivity measurement station.

**Figure 5 materials-16-07455-f005:**
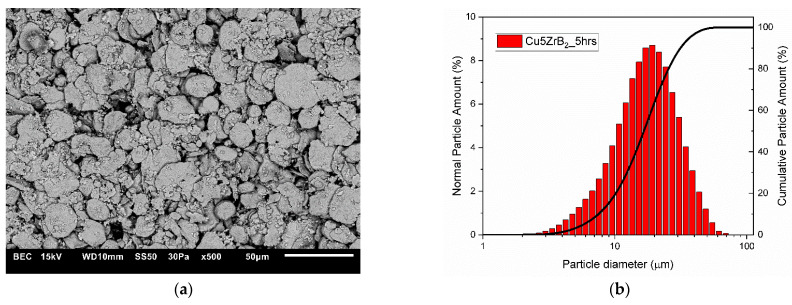
(**a**) Microstructure (SEM) and (**b**) particle size distribution of the powders with 5% ZrB_2_ after 5 h.

**Figure 6 materials-16-07455-f006:**
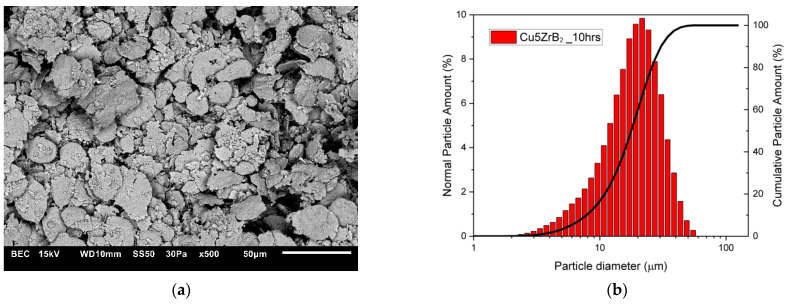
(**a**) Microstructure (SEM) and (**b**) particle size distribution of the powders with 5% ZrB_2_ after 10 h.

**Figure 7 materials-16-07455-f007:**
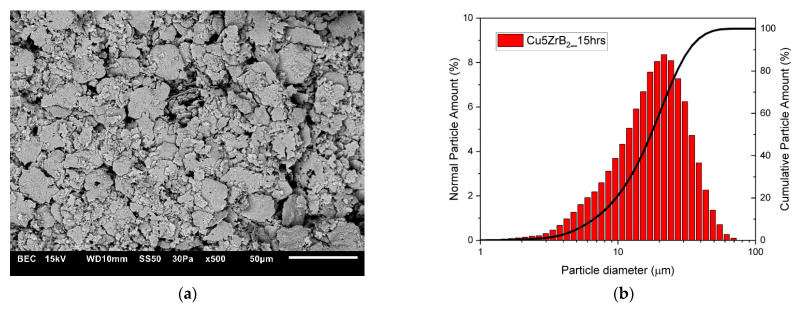
(**a**) Microstructure (SEM) and (**b**) particle size distribution of the powders with 5% ZrB_2_ after 15 h.

**Figure 8 materials-16-07455-f008:**
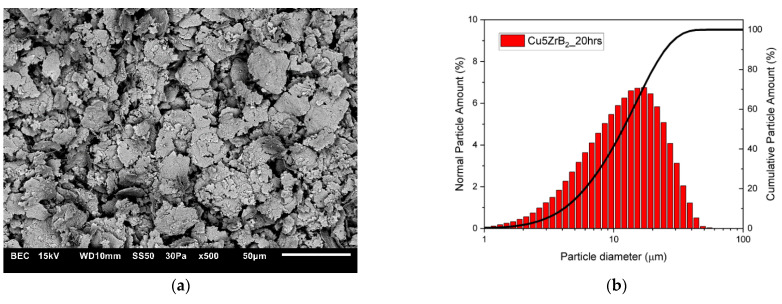
(**a**) Microstructure (SEM) and (**b**) particle size distribution of the powders with 5% ZrB_2_ after 20 h.

**Figure 9 materials-16-07455-f009:**
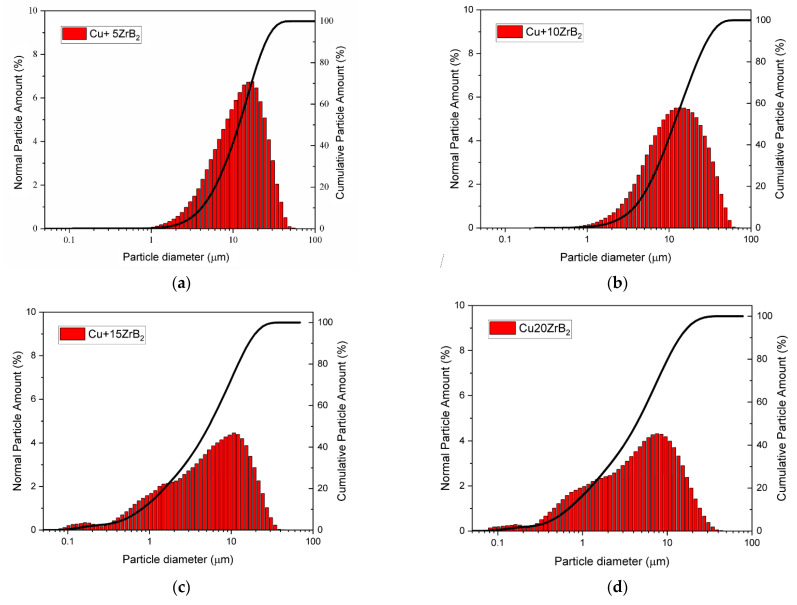
Comparison of the particle size of powders with different ZrB_2_ contents (**a**–**d**).

**Figure 10 materials-16-07455-f010:**
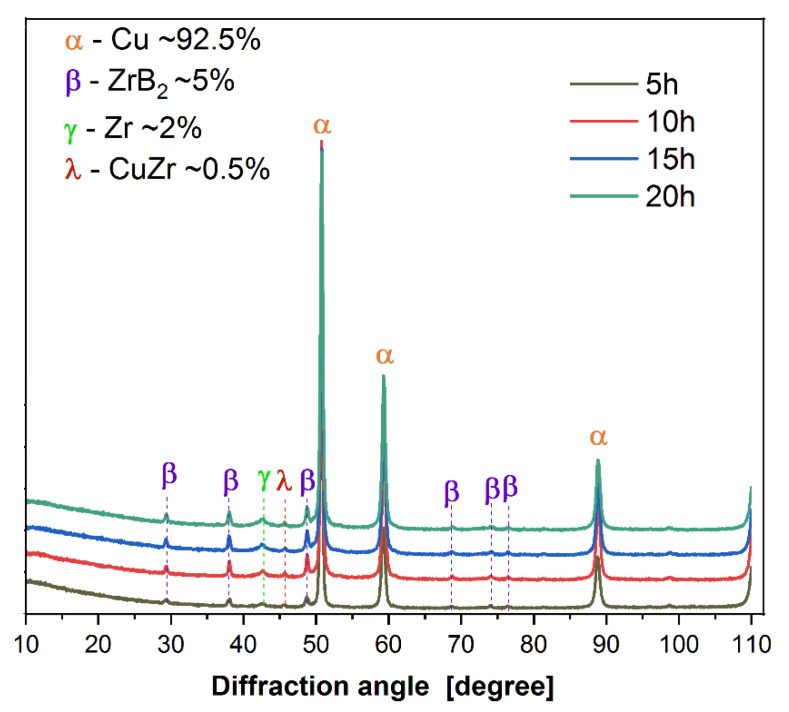
XRD pattern of the Cu + 5ZrB_2_ powders milled for different times.

**Figure 11 materials-16-07455-f011:**
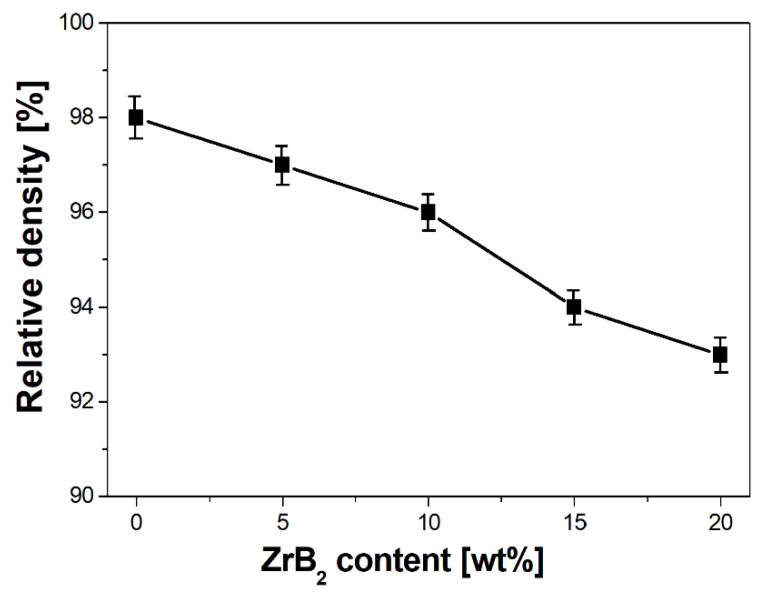
Effect of the ZrB_2_ content on the density of sintered materials (temperature 1223 K).

**Figure 12 materials-16-07455-f012:**
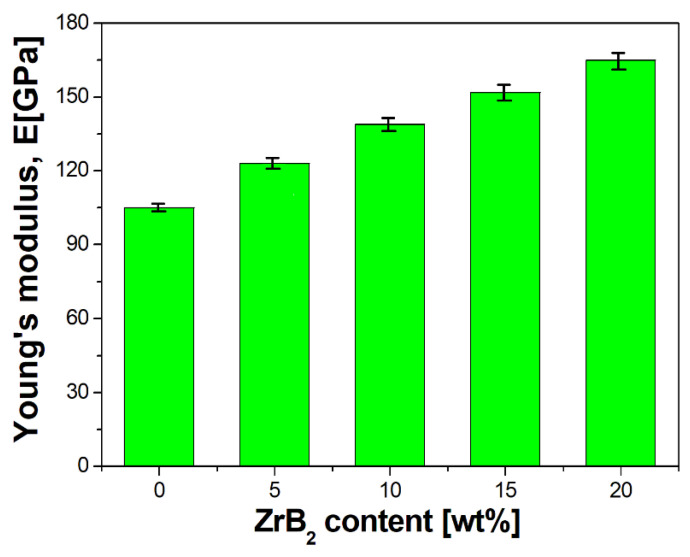
Effect of the ZrB_2_ content on Young’s modulus (temperature 1223 K).

**Figure 13 materials-16-07455-f013:**
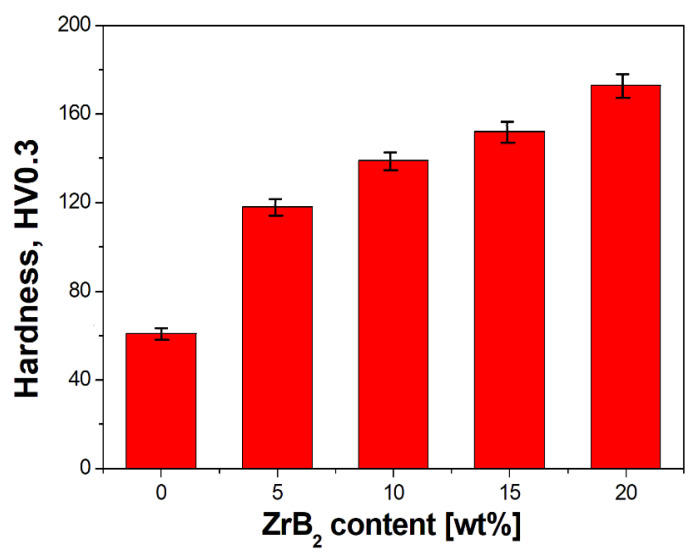
Effect of the ZrB_2_ content on the hardness of sintered materials.

**Figure 14 materials-16-07455-f014:**
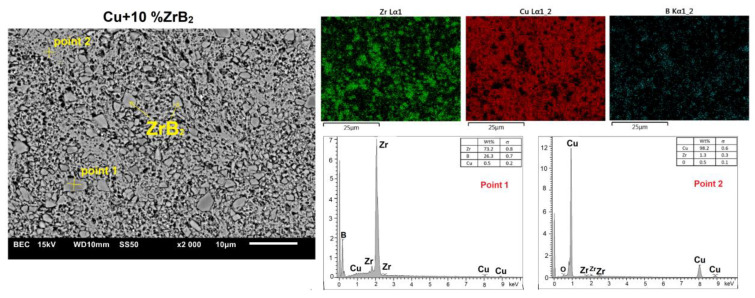
The microstructure (SEM) of Cu + 10%ZrB_2_ composite with the corresponding point analysis and area analysis (WDS).

**Figure 15 materials-16-07455-f015:**
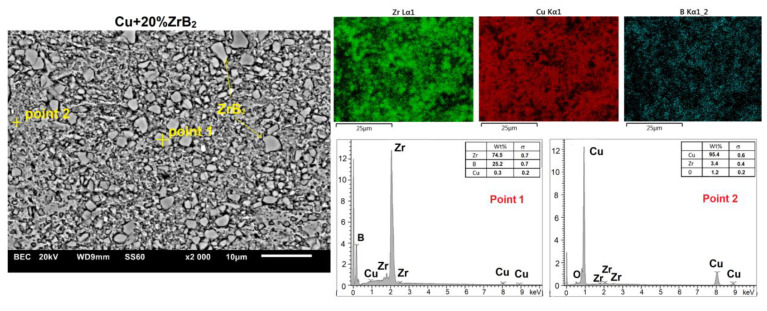
The microstructure (SEM) of Cu + 20%ZrB_2_ composite with the corresponding point analysis and area analysis (WDS).

**Figure 16 materials-16-07455-f016:**
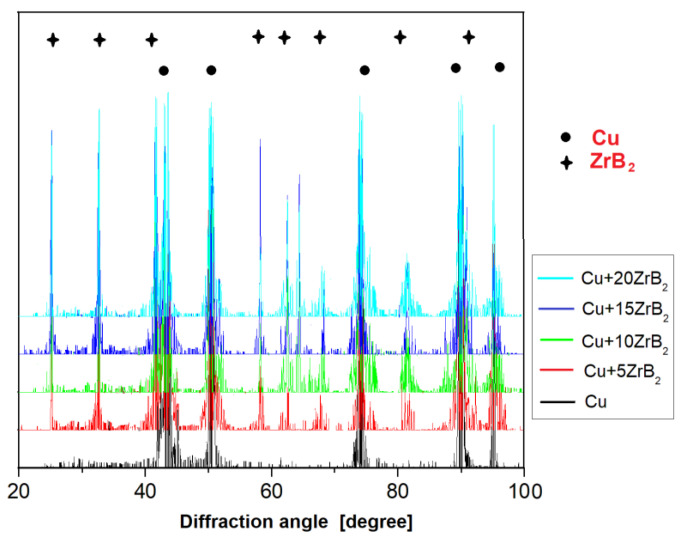
X-ray diffraction patterns of composites after the SPS process at a sintering temperature of 1223 K and a pressure of 35 MPa.

**Figure 17 materials-16-07455-f017:**
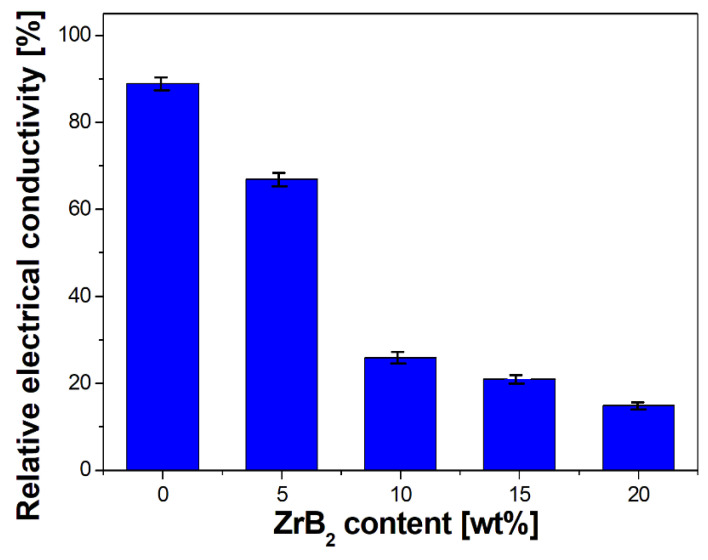
Effect of the ZrB_2_ content on the relative electrical conductivity of sintered materials. A value of 100% conductivity corresponds to pure electrolytic copper, while the bars shown reflect the decrease in electrical performance associated with obtaining composite with different contents of ZrB_2_ (from 0 to 20 wt%) by SPS technology.

**Table 1 materials-16-07455-t001:** The influence of ceramic particles on properties of the Cu-based composites.

Cu-Based Composites	Sintering Condition	Density	Hardness	**Electrical Conductivity**	Ref.
Cu–1 wt% ZrB_2_	Hot-pressed sintering; 840 °C; 25 MPa; 2 h; 10 °C/min	96%	69 HV0.2	96% IACS	[23]
Cu–3 wt% ZrB_2_		95%	84 HV0.2	93% IACS	
Cu–5 wt% ZrB_2_		92.5%	93 HV0.2	88% IACS	
Cu–7 wt% ZrB_2_		91.8%	100.8 HV0.2	83% IACS	
Cu–9 wt% ZrB_2_		91.3%	82 HV0.2	58% IACS	
Cu–5 wt% ZrB_2_	Hot-pressed sintering; 760, 800, 840, 880, 920 °C; 25 MPa; 2 h; 10 °C/min	83–94%	80–92 HV0.2	85–93% IACS	[31]
Cu–2 wt% Mo_2_C	Hot-pressed sintering; 880 °C; 20 MPa; 10 min	91.5%	58.9 HV	83.5% IACS	[32]
Cu–5 wt% Mo_2_C		91.2%	65.8 HV	77.1% IACS	
Cu–7 wt% Mo_2_C		91.3%	69.6 HV	74.7% IACS	
Cu–5%vol TiC	SPS; 800 °C; 10–80 MPa; 5 min	7.0–8.6 g/cm^3^	125-268 HV1	30-53% ISCS	[33]
Cu–1%vol Al_2_O_3_	SPS; 700 °C; 10–50 MPa; 5 min; 80 °C/min	93.2%	77 HV0.3	---	[34]
Cu–5%vol Al_2_O_3_		92.8%	125 HV0.3	---	
Cu–7%vol Al_2_O_3_		86.1%	75 HV0.3	---	

**Table 2 materials-16-07455-t002:** Mechanical milling process parameters.

Mass ratio of ball mass/powder mass	5:1
Material of milling balls	WC (tungsten carbide)
Diameter of milling balls	10 mm
Total milling time	20 h
Milling time/cooling time in one cycle	20 min/10 min
Rotational speeds	200 rpm

**Table 3 materials-16-07455-t003:** Results of the particle size distribution of powders after different milling times.

Powders	Median D (μm)	Modal D (μm)
Cu + 5ZrB_2_ (5 h)	16.587	17.138
Cu + 5ZrB_2_ (10 h)	17.677	21.633
Cu + 5ZrB_2_ (15 h)	17.304	20.952
Cu5 + ZrB_2_ (20 h)	11.752	13.277

**Table 4 materials-16-07455-t004:** Results of particle size distribution of powders with different ZrB_2_ contents.

Powders	Median D (μm)	Modal D (μm)
Cu + 5ZrB_2_	11.752	13.277
Cu + 10ZrB_2_	10.308	11.556
Cu + 15ZrB_2_	4.454	6.750
Cu5 + 20ZrB_2_	3.402	5.348

**Table 5 materials-16-07455-t005:** Influence of the sintering pressure on the properties of the materials formed by SPS.

Sintered Materials	Temperature [K]	Pressure [MPa]	Apparent Density [g/cm^3^]	Relative Density [%]	Open Porosity [%]	Young’s Modulus [GPa]	Relative Young’s Modulus [%]
Cu + 5%ZrB_2_	1123	20	7.45	84	11.34	88	68
	1173		7.68	87	9.89	92	72
	1223		7.89	89	6.95	106	82
	1123	35	8.35	95	2.54	113	88
	1173		8.46	96	1.78	117	91
	1223		8.52	97	1.12	123	97

**Table 6 materials-16-07455-t006:** Influence of the sintering temperature on the properties of the materials formed by SPS.

Sintered Materials	Temperature [K]	Apparent Density [g/cm^3^]	Relative Density [%]	Open Prosity [%]	Young’s Modulus [GPa]	Relative Young’s Modulus [%]
Cu	1123	8.75	98	0.12	107	97
	1173	8.74	98	0.13	107	97
	1223	8.76	98	0.12	105	98
Cu + 5%ZrB_2_	1123	8.35	95	2.54	113	88
	1173	8.46	96	1.78	117	91
	1223	8.52	97	1.12	123	97
Cu + 10%ZrB_2_	1123	7.62	88	9.78	112	76
	1173	7.94	92	3.18	126	85
	1223	8.18	94	1.89	139	96
Cu + 15%ZrB_2_	1123	7.41	87	10.67	119	71
	1173	7.53	88	8.71	132	79
	1223	7.86	92	3.21	154	94
Cu + 20%ZrB_2_	1123	7.22	86	10.54	129	72
	1173	7.31	87	9.73	143	80
	1223	7.57	90	3.83	165	93

## Data Availability

The data presented in this study are available on request from the corresponding author.

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
