# Peer review of "Processing and Properties of ZrB2-Copper Matrix Composites Produced by Ball Milling and Spark Plasma Sintering"

_materials, 2023, doi:10.3390/ma16237455_

Round 1
Reviewer 1 Report
Comments and Suggestions for Authors
This manuscript is devoted to studying the processing and properties copper matrix composites produced by ball milling and spark plasma sintering. The manuscript is well-written and well organized and includes some merits and can be accepted for publication after incorporating the following suggestions:
1. Add some quantitative results in abstract.
2. Add some recent literature on spark plasma sintering of metal matrix composites in the introduction section.
3. Add novelty of present work in the introduction section.
4. How did you decide the range (5 wt% to 20 wt%) of ZrB2 content?
5. Discuss the cooling process of the sintered composite samples in section 2.2
6. Provide suitable references for the measurement of the electrical conductivity of composite samples using four four-point method.
7. Add scientific reason behind the increased hardness of composite sample with an increase in ZrB2 content.
8. Add future scope of the present work in the conclusion section.
Comments on the Quality of English LanguageMinor editing of English language required.
Author Response
Reviewer 1
Comments and Suggestions for Authors
This manuscript is devoted to studying the processing and properties copper matrix composites produced by ball milling and spark plasma sintering. The manuscript is well-written and well organized and includes some merits and can be accepted for publication after incorporating the following suggestions:
- Add some quantitative results in abstract.
Information added in the abstract.
- Add some recent literature on spark plasma sintering of metal matrix composites in the introduction section.
Recent literature added.
- Add novelty of present work in the introduction section.
Information added in the abstract.
The literature analysis showed that there are no data on research on Cu-ZrB2 composite materials sintered using the SPS method. Such composites have been produced using other powder metallurgy technologies [1-3]. Available scientific publications also cover the sintering of copper-based composite materials with various amounts of reinforcing phases, including the following: SiC, NbC, TiC, TiB2, Al2O3 and LaB6-TiB2 [4-9]. Therefore, the novelty of the work is the development and optimisation of the conditions for grinding composite powders and sintering parameters using the SPS method (temperature and pressure) along with the characterisation of selected properties of these materials.
- Ružić, J. Stašić, S. Marković, K. Raić, D. Božić, Synthesis and Characterization of Cu-ZrB2 Alloy Produced by PM Techniques, Science of Sintering, 46 (2014) 217-224
- Zhang Peng, Wang Chenchen, Zhou Shengfeng, Guo Baisong, Zhang Zhiguo, Yu Zhentao, Li Wei, Effect of sintering temperature of the microstructure and properties of high-strength and highly conductive 5 wt.% ZrB2/Cu composite, Powder Metallurgy and Metal Ceramics, 61 (2023)pp. 9-10
- Wang, H. Lin,Z. Zhang W. Li, Fabrication, Interfacial characteristics and strengthening mechanisms of ZrB2microparticles reinforced Cu composites prepared by hot-pressed sintering, Journal of Alloys and Compounds, 748, (2018) 546 - 552
- Celebi Efe G., Altinsoy I., Yener T., Ipek M., Zeytin S., Bindal C.; Characterization of cemented Cu matrix composites reinforced with SiC. Vacuum, 2010, vol. 85, s. 643-647
- Dash K., Ray B. C., Chaira D.; Synthesis and characterisation of copper–alumina metal matrix composite by conventional and spark plasma sintering. Journal of Alloys and Compounds, 2012, vol. 516, s. 78– 84
- Wang F., Li Y., Wang X., Koizumi Y., Kenta Y., Chiba A.; In-situ fabrication and characterization of ultrafine structured Cu-TiC composites with high strength and high conductivity by mechanical milling. Journal of Alloys and Compounds, 2016, vol. 657, s. 122-132
- Soloviova T., Solodkyi I., Loboda P. I.; Spark plasma sintering of Cu-(LaB6-TiB2) metal-ceramic composite and its physical-mechanical properties. Journal of Superhard Materials, 2019, vol. 41, nr 4, s. 213-220
- Long B. D., Othman R., Umemoto M., Zuhailawati H.; Spark plasma sintering of mechanically alloyed in situ copper–niobium carbide composite. Journal of Alloys and Compounds, 2010, vol. 505, s. 510–515
- RuŽić, J. , StaŠić, J. , Rajković, V., Strengthening effects in precipitation and dispersion hardened powder metallurgy copper alloys, Materials and Design, 2013, 49, 746 - 754
- How did you decide the range (5 wt% to 20 wt%) of ZrB2 content?
The paper presents the results of preliminary tests of copper-based composites for which four different weight fractions of ZrB2 (5, 10, 15 & 20 wt.%) were used. No previous research has been conducted to optimize the amount of the reinforcing phase in such composites. At this stage of the research, the assumed weight share of ZrB2 was intended to optimize and select the amount of the reinforcing phase that allows to obtain the best properties of the composites. The results showed that the addition of the reinforcing phase in an amount greater than 10 %wt negatively affects the properties of the copper matrix composites. Therefore, in the further stages of the research, the authors plan to limit themselves to amounts in the range of 0-10% ZrB2 and to produce and test composites containing 2, 4, 6, 8% wt of ZrB2.
- Discuss the cooling process of the sintered composite samples in Section 2.
The sample was cooled at a rate of 373 K / min.
- Provide suitable references for the measurement of the electrical conductivity of composite samples using four four-point method.
The references were added.
- Add scientific reason behind the increased hardness of composite sample with an increase in ZrB2
With increasing content of ZrB2 in the copper matrix, microhardness increases. This is the result of the presence of the ZrB2 reinforcement phase in the copper matrix, which has a very high hardness (>2200 HV [A]). The improvement in microhardness can be explained by the uniform distribution of ZrB2 particles within the Cu matrix. Wang et al. [B] were reported to have increased the microhardness of pure copper (57.5 HV0.2) was increased to 100.8 HV0.2 with an increase in ZrB2 content up to 7 wt%.
[A] Fahrenholtz W.G., Hilmas G.E., Tamy, I.G., Zaykoski J.A. Refractory Diborides of Zirconium and Hafnium. J. Am. Ceram. Soc. 2007;90:1347–1364
[B]. Ch. Wang, H. Lin, Z. Zhang, W. Li, Fabrication, Interfacial characteristics and strengthening mechanisms of ZrB2 microparticles reinforced Cu composites prepared by hot-pressed sintering, Journal of Alloys and Compounds 748 (2018) 546-552
- Add future scope of the present work in the conclusion section.
Currently, detailed microstructural, mechanical, tribological, and corrosion properties are carried out for composites produced under optimal sintering conditions (temperature 950°C, pressure 35 MPa). The results of this research are the subject of a new article. The microstructural characteristics and functional properties obtained will allow us to propose the use of sintered Cu-ZrB2 composites.
Comments on the Quality of English Language
Minor editing of English language required.
The article was again reviewed by native speaker.
Reviewer 2 Report
Comments and Suggestions for Authors
Processing and properties copper matrix composites produced 2 by ball milling and Spark Plasma Sintering
Iwona Sulima and Grzegorz Boczkal
The paper deals with the synthesis and characterization of copper matrix composites with diboride zirconium (ZrB2) by ball milling and consolidated by Spark Plasma Sintering and the effect of the sintering temperature and pressure on the density, porosity and Young’s modulus. It is an interesting technical work, well written and only minor corrections are necessary.
1. Tab 1 file 4, “miling” should be “milling”, please differentiate the milling time and the total milling time indicated in the Table (add the number of milling cycles of 20 min).
2. Indicate the meaning of the all acronyms, particularly the corresponding to non international units (e.g HV or Vickers pyramide hardness…).
3. Explain more clearly the novelties of the work with respect to previous results such as those of the reference 4 (Have no more works been found on Cu-ZrB2 composites?
4. I suggest the addition of ZrB2 on the title (Processing and properties ZrB2-copper matrix composites produced 2 by ball milling and Spark Plasma Sintering).
Comments on the Quality of English LanguageMinor corrections.
Author Response
Reviewer 2
Processing and properties of copper matrix composites produced 2 by ball milling and Spark Plasma Sintering
Iwona Sulima and Grzegorz Boczkal
The paper deals with the synthesis and characterization of copper matrix composites with diboride zirconium (ZrB2) by ball milling and consolidated by Spark Plasma Sintering and the effect of the sintering temperature and pressure on the density, porosity, and Young’s modulus. It is an interesting technical work, well-written, and only minor corrections are necessary.
- Tab 1 file 4, “miling” should be “milling”, please differentiate the milling time and the total milling time indicated in the Table (add the number of milling cycles of 20 min).
It was changed.
- Indicate the meaning of the all acronyms, particularly the corresponding to non international units (e.g HV or Vickers pyramide hardness…)
It was changed.
- Explain more clearly the novelties of the work with respect to previous results such as those of the reference 4 (Have no more works been works found on Cu-ZrB2 composites?
The literature analysis showed that there are no data on research on Cu-ZrB2 composite materials sintered using the SPS method. Such composites have been produced using other powder metallurgy technologies [1-3]. Available scientific publications also cover the sintering of copper-based composite materials with various amounts of reinforcing phases, including the following: SiC, NbC, TiC, TiB2, Al2O3, and LaB6-TiB2 [4-9]. Therefore, the novelty of the work is the development and optimization of the conditions for grinding composite powders and sintering parameters using the SPS method (temperature and pressure) along with the of selected properties of these materials.
- Ružić, J. Stašić, S. Marković, K. Raić, D. Božić, Synthesis and Characterization of Cu-ZrB2 Alloy Produced by PM Techniques, Science of Sintering, 46 (2014) 217-224
- Zhang Peng, Wang Chenchen, Zhou Shengfeng, Guo Baisong, Zhang Zhiguo, Yu Zhentao, Li Wei, Effect of sintering temperature of the microstructure and properties of high-strength and highly conductive 5 wt.% ZrB2/Cu composite, Powder Metallurgy and Metal Ceramics, 61 (2023)pp. 9-10
- Wang, H. Lin,Z. Zhang W. Li, Fabrication, Interfacial characteristics and strengthening mechanisms of ZrB2microparticles reinforced Cu composites prepared by hot-pressed sintering, Journal of Alloys and Compounds, 748, (2018) 546 - 552
- Celebi Efe G., Altinsoy I., Yener T., Ipek M., Zeytin S., Bindal C.; Characterization of cemented Cu matrix composites reinforced with SiC. Vacuum, 2010, vol. 85, s. 643-647
- Dash K., Ray B. C., Chaira D.; Synthesis and characterisation of copper–alumina metal matrix composite by conventional and spark plasma sintering. Journal of Alloys and Compounds, 2012, vol. 516, s. 78– 84
- Wang F., Li Y., Wang X., Koizumi Y., Kenta Y., Chiba A.; In-situ fabrication and characterization of ultrafine structured Cu-TiC composites with high strength and high conductivity by mechanical milling. Journal of Alloys and Compounds, 2016, vol. 657, s. 122-132
- Soloviova T., Solodkyi I., Loboda P. I.; Spark plasma sintering of Cu-(LaB6-TiB2) metal-ceramic composite and its physical-mechanical properties. Journal of Superhard Materials, 2019, vol. 41, nr 4, s. 213-220
- Long B. D., Othman R., Umemoto M., Zuhailawati H.; Spark plasma sintering of mechanically alloyed in situ copper–niobium carbide composite. Journal of Alloys and Compounds, 2010, vol. 505, s. 510–515
- RuŽić, J. , StaŠić, J. , Rajković, V., Strengthening effects in precipitation and dispersion hardened powder metallurgy copper alloys, Materials and Design, 2013, 49, 746 - 754
- I suggest the addition of ZrB2 on the title (Processing and properties ZrB2-copper matrix composites produced 2 by ball milling and Spark Plasma Sintering).
It was changed.
Reviewer 3 Report
Comments and Suggestions for Authors
The paper on Processing and properties copper matrix composites produced by ball milling and Spark Plasma Sintering, by I. Sulima and G. Boczkal, presents a sound analysis of the main characteristics of the copper composite made with diboride zirconium (ZrB2) synthesised by the mentioned process.
The paper is focused into the detailed description of the process and analysis done: shape and size of powder particles, density-porosity, Young's modulus, hardness, and electrical conductivity. The analysis is done considering different temperatures and pressure, thus discussing the effects of the production parameters into the measured properties.
The paper is correctly written, with high-quality pictures and plots to present the results. The discussion presented is soundly supported by the results. The paper clearly deserves publication. Before publication, the following points must be considered by the Authors. Please note that all points refer to some missing information or presentation format, and there is no major criticism about the content.
- - - - - -
As it is clear from the Abstract (L11... with diboride zirconium (ZrB2) were synthesised), the Introduction (L73 ...In the present study, SPS technology was used to consolidation of the Cu composites reinforced with ZrB2...), and repeated in the Conclusions (... ZrB2 content ... addition of 20wt% ZrB2 ... 5 and 10%wt ZrB2...), the copper composites presented are only of one type, reinforced with ZrB2. Therefore, this should be clearly set in the Title of the paper, avoiding any misleading reading about 'whatever copper composites'.
L28
... among others. in the aviation
Remove dot
L82 Kamb Import Export
L83 H.C. Starck
L100 Fritsch Gmb, Germany
L111 XRD, Empyrean PANalytical
L130 Radom, Poland L142 Keithley 2182A
142 nanovoltmeter
L143 Array Electronic 3644A power supply
Provide full company details, similar as for (L87) ...Shimadzu Corporation, Kyoto, Japan.
Table-1
In row 'milling balls', what is WC ?
In row 'Milling time/cooling time' is set 20 min/10 min.
However, the text (L102) mentions 20 minutes of milling - 5 minutes of break.
L121
Figure 3 shows the selected sintering parameters that were registered during SPS process.
Complete the information in the sentence, kind of:
Figure 3 shows the selected sintering parameters that were registered during the SPS process, as function of the process time: Temperature, Force and Piston.
Now, define briefly what is Force and Piston.
L137
X-ray diffraction analysis was carried out using an EMPYREAN diffractometer equipped with a Cu/Kα radiation (Panalytical, The Netherlands).
The equipment was mentioned earlier. Therefore, do not repeat. Consider something as:
X-ray diffraction analysis was carried out using the diffractometer equipped with Cu/Kα radiation.
L153
...the SALD-7500nano ana-153 lyser.
The equipment was mentioned earlier. Therefore, do not repeat. Consider something as:
...the nano analyser.
The paragraph L263-ff can be better introduced with the last line in the previous paragraph: (L256 Analysis of the results indicated ...the ZrB2 reinforcing phase (Figs. 11, 12, Table 5).
Consider moving that sentence to start the next paragraph.
L275
The hardness also showed...
This sentence should start a new paragraph, presenting the new measurement.
Figure 15
Check then caption Cu-10 %wt ZrB2 ?
Equation 2
Check the format of the terms
Comments on the Quality of English Language
The paper is correctly written with a proper flow helping in the reading.
Author Response
Reviewer 3
The paper on Processing and properties copper matrix composites produced by ball milling and Spark Plasma Sintering, by I. Sulima and G. Boczkal, presents a sound analysis of the main characteristics of the copper composite made with diboride zirconium (ZrB2) synthesised by the mentioned process.
The paper is focused into the detailed description of the process and analysis done: shape and size of powder particles, density-porosity, Young's modulus, hardness, and electrical conductivity. The analysis is done considering different temperatures and pressure, thus discussing the effects of the production parameters into the measured properties.
The paper is correctly written, with high-quality pictures and plots to present the results. The discussion presented is soundly supported by the results. The paper clearly deserves publication. Before publication, the following points must be considered by the Authors. Please note that all points refer to some missing information or presentation format, and there is no major criticism about the content.
- - - - -
As itis clear from the Abstract (L11... with diboride zirconium (ZrB2) were synthesised), the Introduction (L73 ...In the present study, SPS technology was used to consolidation ofconsolidate the Cu composites reinforced with repeated in the Conclusions (... ZrB2 content ... addition of 20wt% ZrB2 ... 5 and 10%wt ZrB2...), the copper composites presented are only of one type, reinforced with ZrB2. Therefore, this should be clearly set in the Title of the paper, avoiding any misleading reading about 'whatever copper composites'.
The title of the article has been modified. ‘Processing and properties of ZrB2-copper matrix composites produced 2 by ball milling and Spark Plasma Sintering”
L28
... among others. in the aviation
Remove dot
L82 Kamb Import Export
L83 H.C. Starck
L100 Fritsch Gmb, Germany
L111 XRD, Empyrean PANalytical
L130 Radom, Poland L142 Keithley 2182A
142 nanovoltmeter
L143 Array Electronic 3644A power supply
Provide full company details, similar as for (L87) ...Shimadzu Corporation, Kyoto, Japan.
Table-1
In row 'milling balls', what is WC ?
WC - tungsten carbide
In row 'Milling time/cooling time' is set 20 min/10 min.
However, the text (L102) mentions 20 minutes of milling - 5 minutes of break.
It should be correct: Milling time 20 min; cooling time - 10 min
Missing information has been changed or added to the above detailed comments.
L121
Figure 3 shows the selected sintering parameters that were registered during the SPS process.
Complete the information in the sentence, kind of:
Figure 3 shows the selected sintering parameters that were registered during the SPS process, as a function of the process time: Temperature, Force and Piston.
The sentence was changed.
Now, define briefly what is Force and Piston.
In Figure 3c, the y axis was incorrectly described. The correct text should be: punch displacement
L137
X-ray diffraction analysis was carried out using an EMPYREAN diffractometer equipped with a Cu/Kα radiation (Panalytical, The Netherlands).
The equipment was mentioned earlier. Therefore, do not repeat. Consider something as:
X-ray diffraction analysis was carried out using the diffractometer equipped with Cu/Kα radiation.
It was changed.
L153
...the SALD-7500nano analyser.
The equipment was mentioned earlier. Therefore, do not repeat. Consider something as:
...the nano analyser.
I was corrected.
The paragraph L263-ff can be better introduced with the last line in the previous paragraph: (L256 Analysis of the results indicated ...the ZrB2 reinforcing phase (Figs. 11, 12, Table 5).
Consider moving that sentence to start the next paragraph.
I was corrected.
L275
The hardness also showed...
This sentence should start a new paragraph, presenting the new measurement.
It was changed.
Figure 15
Check then caption Cu-10 %wt ZrB2 ?
It was changed. The correct text should be: Figure 15. The microstructure (SEM) of the Cu + 20% ZrB2 compound with the corresponding point analysis and area analysis (WDS).
Equation 2
Check the format of the terms
It was changed.
Comments on the Quality of English Language
The paper is correctly written with a proper flow helping in the reading.
The article was again reviewed by native speaker
Reviewer 4 Report
Comments and Suggestions for Authors
The authors in the present manuscript show the copper matrix composites with diboride zirconium (ZrB2) were synthesised by ball milling and consolidated by Spark Plasma Sintering (SPS). The characterisations of the ball-milled composite powders were performed by scanning electron microscopy (SEM), X-ray diffraction, and measurement of particle size distribution. The effect of the sintering temperature (1123 K, 1173 K and the 1223 K) and pressure (20 MPa and 35 MPa) on the density, porosity and Young’s modulus were investigated. The relationship between the change of ZrB2 content and physical, mechanical, and electrical properties was studied. Experimental data showed that the properties of Cu-ZrB2 composites depend significantly on the SPS sintering conditions. The optimal sintering temperature is 1223 K and a pressure of 35 MPa. Composites exhibit a high degree of consolidation. For these materials, the apparent density was in the range of 93-97%. The results showed that the higher content of ZrB2 in the copper matrix is responsible for the improvement in Young's modulus and hardness with the reduction of the conductivity of sintered composites. The authors should address the following issues and information’s before publication acceptance in the prestigious ‘Materials’ Journal:
1. In Introduction, authors should add a Table that compares the Cu composites, fabrication parameters and properties with published literatures? Authors should mention some applications of Cu composites.
2. In introduction, can authors explain that what is the novelty of this study?
3. In Experimental, how do authors decide different compositions of ZrB2 (5, 10, 15 & 20wt.%) with Cu composites? Are authors optimized these compositions?
4. In Experimental, what is the impact of this high heating rate (473 K/min) on the composite properties?
5. In Research results, Section 3.1, Why a decrease in the average particle size of the powders was observed with an increase in the amount of ZrB2 after 20 hours of milling?
6. In Figure 14, authors should incorporate the references to support the SEM and EDX analysis of the samples. Authors may go through these two publications for more details and cite accordingly: https://doi.org/10.1016/j.matchemphys.2019.122102 & https://doi.org/10.1016/j.micromeso.2017.08.047
Comments on the Quality of English LanguageMinor editing of English language required.
Author Response
Reviewer 4
The authors in the present manuscript show the copper matrix composites with diboride zirconium (ZrB2) were synthesised by ball milling and consolidated by Spark Plasma Sintering (SPS). The characterisations of the ball-milled composite powders were performed by scanning electron microscopy (SEM), X-ray diffraction, and measurement of particle size distribution. The effect of the sintering temperature (1123 K, 1173 K and the 1223 K) and pressure (20 MPa and 35 MPa) on the density, porosity and Young’s modulus were investigated. The relationship between the change of ZrB2 content and physical, mechanical, and electrical properties was studied. Experimental data showed that the properties of Cu-ZrB2 composites depend significantly on the SPS sintering conditions. The optimal sintering temperature is 1223 K and a pressure of 35 MPa. Composites exhibit a high degree of consolidation. For these materials, the apparent density was in the range of 93-97%. The results showed that the higher content of ZrB2 in the copper matrix is responsible for the improvement in Young's modulus and hardness with the reduction of the conductivity of sintered composites. The authors should address the following issues and information’s before publication acceptance in the prestigious ‘Materials’ Journal:
- In Introduction, authors should add a Table that compares the Cu composites, fabrication parameters and properties with published literatures? Authors should mention some applications of Cu composites.
Information added in the abstract
- In introduction, can authors explain that what is the novelty of this study?
Information added in the abstract.
The literature analysis showed that there are no data on research on Cu-ZrB2 composite materials sintered using the SPS method. Such composites have been produced using other powder metallurgy technologies [1-3]. Available scientific publications also cover the sintering of copper-based composite materials with various amounts of reinforcing phases, including the following: SiC, NbC, TiC, TiB2, Al2O3, and LaB6-TiB2 [4-9]. Therefore, the novelty of the work is the development and optimization of the conditions for grinding composite powders and sintering parameters using the SPS method (temperature and pressure) along with the of selected properties of these materials.
- Ružić, J. Stašić, S. Marković, K. Raić, D. Božić, Synthesis and Characterization of Cu-ZrB2 Alloy Produced by PM Techniques, Science of Sintering, 46 (2014) 217-224
- Zhang Peng, Wang Chenchen, Zhou Shengfeng, Guo Baisong, Zhang Zhiguo, Yu Zhentao, Li Wei, Effect of sintering temperature of the microstructure and properties of high-strength and highly conductive 5 wt.% ZrB2/Cu composite, Powder Metallurgy and Metal Ceramics, 61 (2023)pp. 9-10
- Wang, H. Lin,Z. Zhang W. Li, Fabrication, Interfacial characteristics and strengthening mechanisms of ZrB2microparticles reinforced Cu composites prepared by hot-pressed sintering, Journal of Alloys and Compounds, 748, (2018) 546 - 552
- Celebi Efe G., Altinsoy I., Yener T., Ipek M., Zeytin S., Bindal C.; Characterization of cemented Cu matrix composites reinforced with SiC. Vacuum, 2010, vol. 85, s. 643-647
- Dash K., Ray B. C., Chaira D.; Synthesis and characterisation of copper–alumina metal matrix composite by conventional and spark plasma sintering. Journal of Alloys and Compounds, 2012, vol. 516, s. 78– 84
- Wang F., Li Y., Wang X., Koizumi Y., Kenta Y., Chiba A.; In-situ fabrication and characterization of ultrafine structured Cu-TiC composites with high strength and high conductivity by mechanical milling. Journal of Alloys and Compounds, 2016, vol. 657, s. 122-132
- Soloviova T., Solodkyi I., Loboda P. I.; Spark plasma sintering of Cu-(LaB6-TiB2) metal-ceramic composite and its physical-mechanical properties. Journal of Superhard Materials, 2019, vol. 41, nr 4, s. 213-220
- Long B. D., Othman R., Umemoto M., Zuhailawati H.; Spark plasma sintering of mechanically alloyed in situ copper–niobium carbide composite. Journal of Alloys and Compounds, 2010, vol. 505, s. 510–515
- RuŽić, J. , StaŠić, J. , Rajković, V., Strengthening effects in precipitation and dispersion hardened powder metallurgy copper alloys, Materials and Design, 2013, 49, 746 - 754
- In Experimental, how do authors decide different compositions of ZrB2 (5, 10, 15 & 20wt.%) with Cu composites? Are authors optimized these compositions?
The paper presents the results of preliminary tests of copper-based composites for which four different weight fractions of ZrB2 (5, 10, 15 & 20 wt.%) were used. No previous research has been conducted to optimise the amount of the reinforcing phase in such composites. At this stage of the research, the assumed weight share of ZrB2 was intended to optimise and select the amount of the reinforcing phase that allows one to obtain the best properties of the composites. The results showed that the addition of the reinforcing phase in an amount greater than 10 %wt negatively affects the properties of the copper matrix composites. Therefore, in the further stages of the research, the authors plan to limit themselves to amounts in the range of 0-10% ZrB2 and to produce and test composites containing 2, 4, 6, 8% wt of ZrB2.
- In Experimental, what is the impact of this high heating rate (473 K/min) on the composite properties?
The same heating rate of 473K was used in the tests. The focus was on research on selected sintering conditions (temperature and pressure) on the properties of Cu-ZrB2 composites (density, porosity, Young's modulus). On the basis of this analysis, the optimal sintering conditions (SPS) for such materials were determined. At this stage of the research, no studies were conducted to determine the influence of the heating rate on the properties of sintered composites. Lower heating rates will promote diffusion and affect grain growth and thus deteriorate the mechanical properties of the composites. According to the research A,B], a higher heating rate has a beneficial effect on the final compaction of the material.
[A] M.S. Staltsov, I.I. Chernov, I.A. Bogachev, B.A. Kalin, E.A. Olevsky, L.J. Lebedeva, A.A. Nikitina, Optimization of mechanical alloying and spark-plasma sintering regimes to obtain ferrite–martensitic ODS steel, Nucl. Mater. Energy. 9 (2016) 360-366
[B] Z. Oksiuta, N. Baluc, Optimization of the chemical composition and manufacturing route for ODS RAF steels for fusion reactor application, Nucl. Fusion, 49 (2009),
- In Research results, Section 3.1, Why a decrease in the average particle size of the powders was observed with an increase in the amount of ZrB2 after 20 hours of milling?
Information added in the Research results.
- In Figure 14, authors should incorporate the references to support the SEM and EDX analysis of the samples. Authors may go through these two publications for more details and cite accordingly: https://doi.org/10.1016/j.matchemphys.2019.122102 & https://doi.org/10.1016/j.micromeso.2017.08.047
It was corrected.
Round 2
Reviewer 3 Report
Comments and Suggestions for Authors
The Reviewer appreciates the positive consideration of the comments. The new version included the recommendations provided by the Reviewer.
Author Response
Dear Reviewer 3
Thank you for accepting the revised version of the article.
Yours faithfully,
Iwona Sulima